# The Relative Importance of Herbage Nutritive Value and Climate in Determining Daily Performance per Cow in a Pasture-Based Dairy Farm

**Federico Duranovich** [1,*], **Nicola Shadbolt** [1], **Ina Draganova** [1], **Nicolas López-Villalobos** [1], **Ian Yule** [2] and **Stephen Morris** [1]

1   School of Agriculture and Environment, College of Sciences, Massey University, Private Bag 11-222, Palmerston North 4442, New Zealand; N.M.Shadbolt@massey.ac.nz (N.S.); I.Draganova@massey.ac.nz (I.D.); N.Lopez-Villalobos@massey.ac.nz (N.L.-V.); S.T.Morris@massey.ac.nz (S.M.)
2   Massey AgriFood Digital Lab, School of Food and Advanced Technology, College of Sciences, Massey University, Private Bag 11-222, Palmerston North 4442, New Zealand; I.J.Yule@massey.ac.nz
*   Correspondence: F.N.Duranovich@massey.ac.nz

**Abstract:** The objective of this study was to assess the relative importance of herbage nutritive value (NV), herbage quantity and climate-related factors in determining daily performance per cow in a pasture-based dairy farm. Data on milk production, live weight, body condition score, weather, herbage NV and herbage quantity were regularly collected from August 2016 to April 2017 and from July 2017 to April 2018 at Dairy 1, Massey University, Palmerston North, New Zealand. Data were analyzed using multiple linear regression. Results indicated herbage NV was of higher relative importance in explaining the variation in performance per cow than herbage quantity and climate factors. The relative importance of the interaction between herbage metabolizable energy (ME) and crude protein (CP) on explaining variation in yields of milk, fat and protein was high ($0.11 \leq R^2 \leq 0.15$). Herbage ME was of high relative importance in determining milk urea and body condition score, while neutral detergent fiber was a key driver of milk urea and liveweight ($0.12 \leq R^2 \leq 0.16$). The quantity of herbage supplied at Dairy 1 might have been high enough to not limit cow performance. Developing feeding strategies aimed at improving the efficiency of cow feeding by exploiting the daily variation in herbage NV to better match supply and demand of nutrients may be useful to improve the overall performance per cow of pasture-based dairy farms.

**Keywords:** herbage nutritive value; herbage quantity; climate; grazing cow performance; pasture-based dairy farming

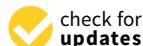

## 1. Introduction

Daily allocation of herbage to cows in New Zealand pasture-based dairy farms has traditionally been focused on monitoring the quantity over the nutritive value (NV) of herbage. The dairy industry encourages the use of herbage quantity measurement tools and ryegrass leaf stage monitoring to promote good grazing management practice [1]. It is assumed that good grazing management would result in little need for herbage NV measurement, as such an approach would anyways result in herbage of optimal NV being offered. However, there is evidence that farmers do not always make optimal grazing management decisions [2]. Moreover, other factors than grazing management, including species composition, soil moisture, soil fertility and climate, also affect herbage NV [3].

Regardless of the cause, variation in the NV of herbage offered daily to dairy cows is likely to exist [4]. Daily variation in herbage NV could, therefore, result in times at which the supply of nutrients do not match the demand for these nutrients. In such situations, the actual performance of cows can differ from that expected by farmers, resulting in inefficient grazing management. It is well known that herbage dry matter intake (DMI),

which is in practice controlled by the allowance of herbage offered, is a major factor determining marginal performance of grazing dairy cows [5–8]. However, climate- [9,10] and herbage-related factors such as herbage NV also play a significant role in influencing cow performance [11,12].

Including a measure of herbage NV to support daily feed allocation has been proposed as an opportunity to improve grazing management efficiency in pasture-based dairy farms [13]. Such inclusion would allow a more precise match between demand and supply of herbage by extending the focus of feed allocation from adequate quantity to adequate nutrition. However, the lack of commercial tools that would allow farmers to rapidly measure herbage NV in the field has most likely contributed to the lack of adoption of such practice. Advances in the field of herbage NV measurement have been made [14,15], with recent work being specifically intended to address this issue for the context of dairy grazing management [16]. For rapid measurement of herbage NV to be useful for farmers, data collected in field-like conditions is required to determine the extent to which daily variation in herbage NV could influence performance per cow in a pasture-based dairy farm. By determining the extent to which herbage NV can drive performance of grazing milking cows in field-like conditions, this study can contribute to the discussion of the importance that should be given to monitoring herbage NV and to the design of feeding strategies that account for the variation in herbage NV. The objective of this study was to determine the relative importance of herbage NV, and other herbage- and climate-related factors on the daily performance of a pasture-based dairy farm on a per cow basis.

## 2. Materials and Methods

### 2.1. Description of the Pasture-Based Dairy Farm

This study was conducted at Dairy 1 (D1) farm, Massey University, Palmerston North, New Zealand (latitude = $-40°$ $22'$ $35.1''$, longitude = $175°$ $36'$ $51.1''$). Dairy 1 is a low-input pasture-based dairy farm system with spring calving and where all the cows in the herd are milked once a day for the full production season.

The climate in the location is temperate, with an annual rainfall of 980 mm, annual temperature of 13.1 °C and low and mean high temperatures of 8.5 and 17.8 °C, respectively [17]. Farm soils comprise a complex assemblage of free-draining alluvial soils including Rangitikei loamy sand, Manawatu fine sandy loam, Manawatu sand loam/gravelly phase, Manawatu mottled silt loam and Karapoti brown sandy loam, with these soils being well drained and naturally fertile. Irrigation is available on nearly 25% of the farm area and is used during summer when soil water deficits are likely to occur.

During the 2016–2017 and 2017–2018 production seasons, the dairy herd had 260 and 255 cows, respectively. Cows were allocated to an effective area of 119.7 ha, which resulted in a stocking rate of about 2.1 cows/ha. The mean stocking density, defined as the instant number of cows per hectare per day, used at Dairy 1 was of 91 (SD = 14) and 119 (SD = 48) cows/ha/day in the 2016–2017 and 2017–2018 production season, respectively. The herd consisted of 25.4% Holstein-Friesian, 22.4% Jersey and 52.4% Holstein-Friesian x Jersey crossbreed. All 65 paddocks in the milking platform have race access, irrigation is available to 35.4 ha, and replacement heifers are grazed off-farm.

The diet offered to cows is mostly composed of home-grown feed. Forage resources grown at D1 are: (1) grass/legume herbage mix (ryegrass/white and red clover) (76% of farm effective area), (2) herb/legume herbage mix (plantain, chicory, white clover, red clover) (12%) and (3) crops (lucerne and maize) (12%). Following excess herbage growth during spring, crops are made into silage or hay and fed to cows at times when feed is in deficit.

### 2.2. Data Collection

The quantity and nutritive value of herbage offered to cows from about four to six paddocks included in the farm manager's weekly grazing plan were measured every two to three weeks during the 2016–2017 (from 1 August 2016 to 9 April 2017) and 2017–2018 (from 31 July 2017 to 8 April 2018) production seasons. Likewise, climate and farm performance

data of the dates in which grazing of paddocks occurred (i.e., grazing event) were collected. The final dataset consisted of observations from 140 days ($n$ = 140; 49 corresponding to the 2016–2017 and 91 to the 2017–2018 production season).

### 2.2.1. Herbage Quantity

Herbage mass (HM) of paddocks at pre-grazing was measured using a C-Dax pasture meter with auto lift (Pasture Meter+ model 5008, C-Dax Agricultural Solutions, Turitea, New Zealand) towed behind an all-terrain vehicle. Runs were made following a "W" shaped pattern across the length of the paddock. Data collected within each paddock were averaged and converted to herbage mass using the following equation calibrated and validated by D1 technical staff:

$$HM \ (kg \ DM/ha) = 752 + 16.3 \times Height \ (mm) \tag{1}$$

At each grazing event, the paddock identification number, date and area allocated (AH) to the milking herd were recorded. When there was more than one grazing event in a day, the mean HM of the herbage on offer for the day was determined by weighting the area of the paddocks allocated to the herd. The amount of herbage on offer on a day-to-day basis was calculated by multiplying HM by AH. Herbage allowance (HA) was calculated by dividing the amount of herbage on offer for the day by the number of grazing cows. The percentage of herbage in the diet of cows (PD) was calculated by dividing HA by the total feed offered to the herd per cow (HA + supplements) and multiplied by 100.

### 2.2.2. Herbage Nutritive Value

Herbage NV traits of metabolizable energy (ME), crude protein (CP) and neutral detergent fiber (NDF) of paddocks were determined from canopy hyperspectral measurements acquired from twelve sampling plots distributed along the runs performed with the pasture meter right after herbage mass measurement. The number of plots was defined following the recommendation of Cosgrove et al. [18] who suggest that twelve samples are required to determine the mean herbage NV of a paddock with accuracies of ±0.5 MJ/kg DM for ME and of ±5% for CP and NDF. The description of the instrument used to acquire spectra, the definition of sampling plot and the accuracy of the calibrations used to determine the different herbage NV traits are detailed in [16]. Similar to HM, when there was more than one paddock being grazed in a day, the mean NV of the herbage on offer for the day was determined by weighting the area of the paddocks allocated to the herd.

### 2.2.3. Climate

Weather data (mean, max and min air temperatures, relative humidity, wind speed and rainfall) were obtained from the Palmerston North Ews station available at the national climate database [17]. Weather data were used to calculate a temperature humidity index [19] and a cold stress index [20] as follows:

$$THI = 0.8 \times T + [RH \times (T_{max} - 14.4)] + 46.4 \tag{2}$$

where THI is temperature humidity index, T is mean daily temperature (°C), $T_{max}$ is maximum daily temperature (°C), and RH is mean daily percent relative humidity divided by 100.

$$CSI = [11.7 + (3.1 \times WS^{0.5})] \times (40 - T) + 481 + 418 \ (1 - e^{-0.04 \times R}) \tag{3}$$

where CSI is cold stress index (kJ/m$^2$/h), WS is mean daily wind speed (m/s), T is mean daily temperature (°C), e is Euler's number (mathematical constant), and R is total daily rainfall (mm).

2.2.4. Pasture-Based Dairy Farm Performance per Cow

Milk production at the farm was monitored using the dairy company actual milk supply records. Records for milk, milk solids (fat + protein), fat, protein and milk urea obtained at the vat were divided by the number of cows milked that day to obtain daily yields of milk (MY), fat (FY), protein (PY), milk solids (MSY) and urea (MUY) per cow. Percentages of fat (FP) and protein (PP), protein to fat ratio (PFR) and urea concentration (MU) of the milk produced at the farm were also expressed per cow. Daily live weights (LW) of cows identified with a radio frequency electronic identification system (Allflex New Zealand Ltd., Palmerston North, New Zealand) were automatically measured every morning after milking using an automatic race walkover scale situated in the exit of the milking shed (WoW xR-3000, Tru-Test Ltd., Auckland, New Zealand). The body condition score (BCS) of all cows in the herd was assessed once every month using a 10-point scale [21] by the Dairy 1 farm research technician. In order to account for missing data and to allow the daily characterization of LW and BCS, these parameters were modelled for each of the cows as a function of their days in milk using Legendre polynomials of 3rd order over the two production seasons [22]. These models were used to generate LW and BCS data for each day in which the cows were present in the milking shed. For each calendar day, LW- and BCS-generated data were averaged in order to obtain single daily values. The average change in live weight per cow (LWC) in the herd was calculated as the difference in LW between successive days. Performance per cow data were paired with herbage and weather data from the day after the grazing event took place.

*2.3. Development of Overall Performance Indices*

Per cow performance variables MY, FY, PY, MSY, MUY, FP, PP, PFR, MU, LW, LWC and BCS were combined into two performance indices by carrying out a principal component analysis (PCA) in RStudio software (R Studio version 1.2.5019, R Team, Boston, MA, USA). Prior to the analysis, variables were scaled to a zero mean and a standard deviation of one. Loadings for the first (PC1) and second (PC2) principal components were interpreted in light of the original variables and were used to describe overall indices of daily performance per cow termed Performance Index 1 (PI1) and Performance Index 2 (PI2), respectively. The sign of the scores were rescaled based on the interpretation of the principal components to a minimum of zero and a maximum of one hundred.

*2.4. Statistical Analysis*

In order to determine the influence of herbage NV, herbage quantity and climate-related factors on daily performance per cow, a multiple linear regression (MLR) modelling approach was implemented using the "caret" package available for RStudio software [23]. Explanatory and dependent modelling variables are detailed in Table 1.

Highly correlated explanatory variables were excluded from the models by setting a cut-off value for pair-wise correlations of 0.9. The MLR algorithm used a step-wise variable selection criteria that selected the model with the lowest Akaike information criterion (AIC) value. Performance of the developed models was assessed by calculating the coefficient of determination ($R^2$) for all, and each one of the explanatory variables included in a model. The proportion of the variance in a response variable that was explained by the variance of an explanatory variable was defined as the "relative importance" of an explanatory variable and was calculated using the method described by Lindeman [24] as implemented in the package "relaimpo" for RStudio [25]. Regression coefficients were used to investigate the relationships between explanatory and response variables.

**Table 1.** Model variable list, description and units.

| Variable Group | | Variable Name | Description | Units |
|---|---|---|---|---|
| X—Explanatory variables | Herbage quantity | HM | herbage mass | kg DM/ha |
| | | HA | herbage allowance | kg DM/cow/day |
| | | AH | area of herbage offered | ha/day |
| | | PD | percentage of herbage in diet | % of dietary DM |
| | Herbage nutritive value | ME | herbage metabolizable energy | MJ/kg DM |
| | | CP | herbage crude protein | %DM |
| | | NDF | herbage neutral detergent fiber | %DM |
| | | DM | herbage dry matter | %FM |
| | | ME×CP | ME and CP interaction | - |
| | Climate | T | mean daily temperature | °C |
| | | THI | temperature humidity index | index |
| | | CSI | cold stress index | kJ/m$^2$/h |
| | | Rain | rainfall | mm/day |
| | Time | POP | period of production defined in days from the beginning of milk production as: Early = days 1 to 90 Mid = days 91 to 180 Late = days 181 to 250 | - |
| | | Y | production season defined as: 2016 = 2016–2017 production season 2017 = 2017–2018 production season | - |
| Y—Dependent variables | | MY | milk yield per cow in the herd | L/cow |
| | | MSY | milk solids yield per cow in the herd | kg MS/cow |
| | | MSP | milk solids percentage per cow in the herd | %MY |
| | | FP | milk fat percentage per cow in the herd | %MY |
| | | PP | milk protein percentage per cow in the herd | %MY |
| | | FY | milk fat yield per cow in the herd | kg F/cow |
| | | PY | milk protein yield per cow in the herd | kg P/cow |
| | | PFR | milk protein to fat ratio per cow in the herd | Ratio |
| | | MU | milk urea concentration per cow in the herd | mg/dL |
| | | MUY | milk urea yield per cow in the herd | kg MU/cow |
| | | LW | live weight per cow in the herd | kg LW/cow |
| | | LWC | live weight change per cow in the herd | kg LW/cow/day |
| | | BCS | body condition score per cow in the herd | index (1–10 scale) |
| | | PI1 | performance index 1 | index (0–100 scale) |
| | | PI2 | performance index 2 | index (0–100 scale) |

FM = herbage fresh matter (kg).

## 3. Results

### 3.1. Overall Performance Indices

Principal components 1 and 2 were accountable for 49 and 18% of the variance of the performance per cow data, respectively. As indicated by the principal component loadings in Table 2, the first performance index resulting from PC1 (PI1) was closely associated with yields of milk, milk solids, milk fat and milk protein. Conversely, the second performance index resulting from PC2 (PI2) was associated with BCS, LW, LWC, milk urea concentration (MU), MUY, FP and MSP.

### 3.2. Descriptive Statistics

Descriptive statistics of explanatory and response modelling variables are presented in Table 3. Performance indices 1 and 2, MU, MUY, and yields for milk, milk solids, protein and fat were the response variables that varied the most (11.1 ≤ CV% ≤ 47.9). The explanatory variables that varied the most were AH, HA, PD and T (25.7 ≤ CV% ≤ 34.7). The coefficient of variation in any of the NV traits measured was relatively low. Among NV traits, DM and CP varied more than ME and NDF.

**Table 2.** Principal component loadings that were used to define overall performance indices 1(PI1) and 2 (PI2) based on the combination of performance per cow variables. Variance is explained by the PI within brackets.

| Variable | PI1 (49%) | PI2 (18%) |
|----------|-----------|-----------|
| MY | −0.39 | 0.04 |
| MSY | −0.37 | 0.13 |
| MSP | 0.31 | 0.35 |
| PFR | 0.06 | −0.11 |
| FP | 0.26 | 0.37 |
| PP | 0.29 | 0.25 |
| FY | −0.36 | 0.14 |
| PY | −0.35 | 0.10 |
| MU | 0.27 | −0.25 |
| MUY | 0.04 | −0.33 |
| LW | 0.24 | 0.34 |
| LWC | 0.11 | −0.40 |
| BCS | −0.23 | 0.39 |

MY = milk yield per cow in the herd, MSY = milk solids yield per cow in the herd, MSP = milk solids percentage per cow in the herd, FP = milk fat percentage per cow in the herd, PP = milk protein percentage per cow in the herd, FY = milk fat yield per cow in the herd, PY = milk protein yield per cow in the herd, PFR = milk protein to fat ratio per cow in the herd, MU = milk urea concentration per cow in the herd, MUY = milk urea yield per cow in the herd, LW = live weight per cow in the herd, LWC = live weight change per cow in the herd, BCS = body condition score per cow in the herd.

**Table 3.** Descriptive statistics of herbage nutritive value, herbage quantity and climate explanatory variables and performance per cow response variables used in this study.

| | Variable | *n* | Mean | SD | CV% | Min | Max |
|---|----------|-----|------|-----|-----|-----|-----|
| X—Explanatory variables | HM (kg DM/ha) | 140 | 2908 | 204 | 7.0 | 2514 | 3487 |
| | HA (kg DM/cow/day) | 140 | 29.5 | 8.54 | 28.9 | 10.89 | 46.28 |
| | AH (ha/day) | 140 | 2.29 | 0.79 | 34.7 | 0.50 | 4.04 |
| | PD (%DM) | 140 | 77.3 | 27.4 | 27.3 | 23.0 | 100.0 |
| | ME (MJ/kg DM) | 140 | 10.9 | 0.52 | 4.7 | 9.36 | 11.66 |
| | CP (%DM) | 140 | 17.5 | 2.15 | 12.3 | 10.77 | 21.89 |
| | NDF (%DM) | 140 | 39.6 | 3.17 | 8.0 | 34.7 | 49.7 |
| | DM (%FM) | 140 | 21.8 | 4.32 | 19.8 | 15.38 | 35.78 |
| | T (°C) | 140 | 15.0 | 3.86 | 25.7 | 6.39 | 23.32 |
| | THI | 140 | 66.2 | 7.02 | 10.6 | 51.69 | 81.04 |
| | CSI (kJ/m$^2$/h) | 140 | 1199 | 78.4 | 6.5 | 1085 | 1495 |
| | Rain (mm/day) | 140 | 2.8 | 4.82 | – | 0.00 | 25.40 |
| Y—Dependent variables | MY (L/cow/day) | 140 | 16.1 | 2.13 | 13.2 | 9.53 | 19.77 |
| | MSY (kg MS/cow/day) | 140 | 1.47 | 0.16 | 11.1 | 0.90 | 1.76 |
| | MSP (%MY) | 140 | 9.06 | 0.38 | 4.2 | 8.57 | 10.47 |
| | FP (%MY) | 140 | 5.20 | 0.23 | 4.5 | 4.77 | 6.00 |
| | PP (%MY) | 140 | 3.96 | 0.19 | 4.9 | 3.63 | 5.51 |
| | FY (kg F/cow) | 140 | 0.83 | 0.09 | 11.4 | 0.52 | 1.00 |
| | PY (kg P/cow) | 140 | 0.63 | 0.07 | 11.2 | 0.38 | 0.78 |
| | PFR | 140 | 0.76 | 0.03 | 4.2 | 0.65 | 0.84 |
| | MU (mg/dL) | 140 | 20.3 | 5.13 | 25.3 | 9.20 | 32.50 |
| | MUY (kg MU/cow) | 140 | 3.19 | 0.65 | 20.4 | 0.03 | 0.13 |
| | LW (kg LW/cow) | 140 | 479.1 | 4.03 | 0.8 | 474.0 | 492 |
| | LWC (kg LW/cow/day) | 140 | 0.03 | 0.41 | – | −1.57 | 1.33 |
| | BCS (1–10 scale) | 140 | 4.61 | 0.18 | 3.8 | 4.42 | 5.02 |
| | PI1 (1–100 scale) | 140 | 68.4 | 23.7 | 34.2 | 0 | 100 |
| | PI2 (1–100 scale) | 140 | 39.9 | 19.1 | 47.9 | 0 | 100 |

HM = herbage mass, HA = herbage allowance, AH = area of herbage offered, PD = percentage of herbage in diet, ME = herbage metabolizable energy, CP = herbage crude protein, NDF = neutral detergent fiber, DM = herbage dry matter, FM = herbage fresh matter, T = mean daily temperature, THI = temperature humidity index, CSI = cold stress index, Rain = rainfall, MY = milk yield per cow in the herd, MSY = milk solids yield per cow in the herd, MSP = milk solids percentage per cow in the herd, FP = milk fat percentage per cow in the herd, PP = milk protein percentage per cow in the herd, FY = milk fat yield per cow in the herd, PY = milk protein yield per cow in the herd, PFR = milk protein to fat ratio per cow in the herd, MU = milk urea concentration per cow in the herd, MUY = milk urea yield per cow in the herd, LW = live weight per cow in the herd, LWC = live weight change per cow in the herd, BCS = body condition score per cow in the herd, PI1 = performance index 1, PI2 = performance index 2. SD = standard deviation, CV% = coefficient of variation in percentage.

### 3.3. Relative Importance of Herbage and Climate Factors on Performance per Cow

The relative importance and regression coefficients of herbage NV, herbage quantity and climate variables that were used to explain the different responses in performance per cow are shown in Tables 4 and 5, respectively. The number of explanatory variables retained by the various models ranged from 5 to 11. Most performance per cow variables were explained with high $R^2$ values ($0.67 \leq R^2 \leq 0.96$), and only FP, PFR, MUY and LWC were explained with relatively lower $R^2$ values ($0.38 \leq R^2 \leq 0.53$).

**Table 4.** Relative importance [1] of herbage nutritive value, herbage quantity and climate variables in explaining responses in performance per cow.

| Explanatory Variable | Dependent Variable | | | | | | | | | | | | | | |
|---|---|---|---|---|---|---|---|---|---|---|---|---|---|---|---|
| | MY | MSY | MSP | PFR | FP | PP | FY | PY | MU | MUY | LW | LWC | BCS | PI1 | PI2 |
| HM | 0.03 | | 0.03 | 0.02 | | 0.04 | 0.04 | | 0.04 | | 0.02 | 0.01 | 0.02 | 0.03 | 0.02 |
| HA | 0.04 | | 0.03 | 0.07 | 0.01 | 0.07 | | | | 0.03 | 0.04 | **0.11** | 0.08 | 0.04 | **0.10** |
| AH | 0.02 | | 0.05 | **0.11** | 0.01 | **0.13** | 0.03 | | 0.02 | 0.03 | 0.05 | **0.24** | 0.07 | 0.02 | **0.12** |
| PD | | | | | | | | | 0.08 | 0.06 | 0.01 | | | 0.03 | |
| ME | | | | | | | | | **0.14** | 0.04 | 0.08 | | **0.12** | | |
| CP | 0.07 | **0.10** | 0.05 | 0.04 | | 0.08 | **0.10** | 0.07 | | 0.04 | 0.02 | 0.04 | 0.07 | 0.07 | |
| ME×CP | 0.09 | **0.15** | 0.06 | 0.04 | | **0.10** | **0.15** | **0.11** | 0.04 | 0.04 | | 0.06 | **0.11** | **0.10** | 0.03 |
| NDF | 0.08 | | 0.03 | 0.02 | 0.01 | 0.05 | | | **0.12** | 0.08 | **0.15** | | | 0.07 | **0.16** |
| DM | | | | 0.03 | | | | | | 0.01 | | | | | |
| T | | | 0.04 | 0.02 | 0.05 | | | | | | | | 0.21 | | **0.16** |
| CSI | 0.02 | | 0.06 | 0.03 | 0.09 | 0.01 | | 0.02 | | | | | | 0.02 | 0.02 |
| Rain | | | | | | | | | | | 0.03 | | | | |
| POP | **0.43** | **0.51** | **0.33** | 0.02 | **0.30** | **0.25** | **0.47** | **0.47** | **0.24** | 0.08 | **0.27** | | **0.28** | **0.45** | 0.08 |
| Y | 0.02 | 0.01 | 0.04 | 0.01 | 0.03 | 0.04 | 0.01 | 0.01 | 0.01 | | | 0.01 | 0.00 | 0.02 | 0.02 |
| Total $R^2$ | 0.80 | 0.77 | 0.72 | 0.38 | 0.53 | 0.78 | 0.80 | 0.69 | 0.70 | 0.41 | 0.67 | 0.47 | 0.96 | 0.85 | 0.71 |

[1] proportion of the variance in a response variable that was explained by the variance of an explanatory variable. Bold numbers indicate explanatory variables with relative importance coefficient $\geq 0.10$. MY = milk yield per cow in the herd, MSY = milk solids yield per cow in the herd, MSP = milk solids percentage per cow in the herd, FP = milk fat percentage per cow in the herd, PP = milk protein percentage per cow in the herd, FY = milk fat yield per cow in the herd, PY = milk protein yield per cow in the herd, PFR = milk protein to fat ratio per cow in the herd, MU = milk urea percentage per cow in the herd, MUY = milk urea yield per cow in the herd, LW = live weight per cow in the herd, LWC = live weight change per cow in the herd, BCS = body condition score per cow in the herd, PI1 = performance index 1, PI2 = performance index 2, HM = herbage mass, HA = herbage allowance, AH = area of herbage offered, PD = percentage of herbage in diet, ME = herbage metabolizable energy, CP = herbage crude protein, NDF = neutral detergent fiber, DM = herbage dry matter, T = mean daily temperature, ME×CP = ME and CP interaction term, CSI = cold stress index, Rain = rainfall, POP = period of production, Y = production season.

The average relative importance of herbage nutritive value and herbage quantity variables in explaining any of the performance per cow dependent variables was higher than the average relative importance of climate variables ($R^2 = 0.34$ vs. $R^2 = 0.05$, respectively). On average, herbage NV variables were of higher relative importance than herbage quantity variables ($R^2 = 0.21$ vs. $R^2 = 0.13$, respectively). Period of production (POP) was able to explain between 2 and 51% of the variation in all performance per cow dependent variables except LWC (Table 4). The early period of production (the first 90 days from the beginning of milk production) was associated with higher MSY, FY, PY, MUY, BCS and PI1, but lower PP, FP, PI2 compared to mid (day 91 to 180) and late (day 181 to 250) periods ($p < 0.1$–$0.001$) (Table 5). The relative importance of the production season in explaining performance per cow was low ($0.01 \leq R^2 \leq 0.04$) (Table 4), and there were significant differences between production seasons ($p < 0.1$–$0.001$) (Table 5).

**Table 5.** Regression coefficients of herbage nutritive value, herbage quantity and climate variables explaining responses in performance per cow.

| Explanatory Variable | Dependent Variable | | | | | | | | | | | | | | |
|---|---|---|---|---|---|---|---|---|---|---|---|---|---|---|---|
| | MY | MSY | MSP | PFR | FP | PP | FY | PY | MU | MUY | LW | LWC | BCS | PI1 | PI2 |
| Intercept | 12.9 *** | 1.6 *** | 11.4 *** | 1.1 *** | 4.6 *** | 5.9 *** | 0.8 *** | 0.8 *** | 45.1 * | 23.6 * | 578 *** | −2.1 *** | 3.4 *** | 8.3 † | 213 *** |
| HM | 0.001 * | | −0.0003 ** | −0.0001 * | | −0.0002 *** | 0.0001 † | | −0.002 † | | −0.003 * | **0.0005** ** | **−0.0001** *** | 0.01 * | −0.02 ** |
| HA | −0.07 * | | 0.03 *** | 0.002 * | 0.01 * | 0.02 *** | | | −0.03 † | | 0.5 *** | **−0.05** *** | 0.01 *** | −1.1 ** | **3.1** *** |
| AH | 0.7 ** | | −0.4 *** | **−0.03** *** | −0.07 † | **−0.28** *** | 0.01 † | | 1.6 ** | 0.47 ** | **−4.6** *** | −0.03 † | **0.7** *** | **8.4** ** | **−32.7** *** |
| PD | | | | | | | | | | −0.07 *** | −0.01 * | −0.03 † | | 0.1 † | |
| ME | | | | | | | | | **−4.05** ** | −2.1 * | **−3.8** *** | | **0.1** * | | |
| CP | −1.1 *** | **−0.08** *** | 0.2 *** | 0.03 *** | | 0.2 *** | −0.05 *** | −0.02 ** | −1.1 * | | −0.5 *** | 0.1 *** | **0.07** † | −14.4 *** | |
| ME×CP | 0.1 *** | **0.006** *** | −0.02 *** | −0.002 *** | | −0.02 *** | 0.004 *** | 0.002 ** | 0.05 ** | 0.1 * | | | **−0.006** † | **1.3** *** | −0.2 ** |
| NDF | 0.09 † | | −0.05 *** | −0.004 * | −0.01 † | −0.04 *** | | | **0.47** ** | 0.08 * | −1.1 *** | | | **1.7** *** | **−3.1** *** |
| DM | | | | | 0.01 * | | | | | | −0.03 † | | | | |
| T | | | −0.03 ** | 0.002 * | −0.02 *** | | | | | | | | **−0.01** *** | | **−1.1** ** |
| CSI | −0.002 † | | 0.001 *** | −0.0001 * | | 0.0002 † | | −0.0001 † | | | | | | −0.03 ** | 0.04 ** |
| Rain | | | | | | | | | | | | 0.1 ** | | | |
| POP:Mid | **−0.9** ** | **−0.07** ** | **0.1** † | 0.004 * | **0.08** † | **0.04** † | −0.04 ** | −0.03 *** | **−2.6** ** | −0.6 *** | **2.6** ** | | **−0.1** *** | −8.2 * | 7.9 * |
| POP:Late | **−3.6** *** | **−0.2** *** | **0.6** *** | −0.01 † | **0.4** *** | **0.2** *** | −0.1 *** | −0.1 *** | 1.9 † | −0.5 * | **8.3** *** | | **−0.2** *** | −39.6 *** | 19.5 *** |
| Y:2017 | 0.8 *** | 0.04 * | −0.2 *** | −0.01 * | −0.05 † | −0.1 *** | 0.03 ** | 0.02 † | −0.9 † | | | 0.1 * | −0.03 *** | 7.5 *** | −5.8 * |

† Significant at $p < 0.1$, * Significant at $p < 0.05$, ** Significant at $p < 0.01$, *** Significant at $p < 0.001$. Bold numbers indicate explanatory variables with relative importance coefficient ≥ 0.10 as shown in Table 4. MY = milk yield per cow in the herd, MSY = milk solids yield per cow in the herd, MSP = milk solids percentage per cow in the herd, FP = milk fat percentage per cow in the herd, PP = milk protein percentage per cow in the herd, FY = milk fat yield per cow in the herd, PY = milk protein yield per cow in the herd, PFR = milk protein to fat ratio per cow in the herd, MU = milk urea concentration per cow in the herd, MUY = milk urea yield per cow in the herd, LW = live weight per cow in the herd, LWC = live weight change per cow in the herd, BCS = body condition score per cow in the herd, PI1 = performance index 1, PI2 = performance index 2, HM = herbage mass, HA = herbage allowance, AH = area of herbage offered, PD = percentage of herbage in diet, ME = herbage metabolizable energy, CP = herbage crude protein, NDF = neutral detergent fiber, DM = herbage dry matter, T = mean daily temperature, ME × CP = ME and CP interaction term, CSI = cold stress index, Rain = rainfall, POP:Mid = period of production defined between day 91 and day 180 from the beginning of milk production, POP:Late = period of production defined between day 181 and day 250 from the beginning of milk production, Y:2017 = 2017–2018 production season.

The interaction between herbage ME and CP (ME×CP) accounted for between 10 and 15% of the variation in MY, MSY, FY, PY and PI1 (Table 4), with the relationships between ME×CP and MY, MSY, FY, PY and PI1 being positive ($p < 0.01$) (Table 5). Herbage CP was of high relative importance in determining MSY and FY ($R^2 = 0.10$), with the relationships between herbage CP and the two variables being negative ($p < 0.001$). Metabolizable energy and NDF content in herbage were of high relative importance in determining MU ($R^2 = 0.14$ and 0.12, respectively) being the relationship between ME and MU and between NDF and MU negative and positive, respectively ($p < 0.01$). Herbage ME was also an important determinant of BCS ($R^2 = 0.12$) with increasing levels of ME being related with increasing BCS ($p < 0.05$). In addition, NDF was of high relative importance in determining LW and PI2 ($R^2 = 0.15$ and 0.16, respectively) with increasing levels of NDF being related to decreasing LW and PI2 ($p < 0.001$).

Among herbage quantity variables, AH was of high relative importance in the determination of LWC, PFR, PI2 and PP ($0.11 \leq R^2 \leq 0.24$), while HA was of high relative importance in determining LWC and PI2 ($R^2 = 0.11$ and 0.10, respectively) (Table 4). Increasing levels of AH were related with higher LWC, but lower PFR, PP and PI2 ($p < 0.001$) (Table 5). Moreover, HA was negatively related to LWC but positively to PI2 ($p < 0.001$). Among climate variables, T was of high relative importance in explaining BCS and PI2 ($R^2 = 0.21$ and 0.16, respectively) with both dependent variables decreasing with increasing T.

## 4. Discussion

This study sought out to determine the relative importance of the nutritive value of herbage and other herbage- and climate-related factors on the performance of a pasture-based dairy farm on a per cow basis. The following sections discuss the findings of this study in light of the existing literature on the topic and their implication to farm management.

### 4.1. Overall Pasture-Based Dairy Farm Performance per Cow

An interesting feature in this research was the development of two performance indices based on a multivariate analysis of the data. Performance index 1 reflected processes associated with daily milk production, which are important determinants of short-term net profit in pasture-based systems [26]. In contrast, PI2 reflected processes that would impact farm profits over a lengthier period of time, as it was closely related with LW, LWC, BCS and MU, variables that can affect reproduction of cows [27,28] and, thus, influence long-term profit or sustainability of the farm system.

### 4.2. Importance of Seasonality on Performance per Cow

The fact that any performance per cow variable was highly related to the period of production can be explained by the seasonal nature of pasture-based dairy farming. Like most dairy farms in New Zealand, Dairy 1 is managed to ensure all cows calve between late winter and early spring and are dried-off in autumn. Such management synchronizes herd feed demand with the seasonal herbage growth pattern, allowing milk production at low cost [29]. This practice also signifies changes in the physiological stage of cows that result in temporal variation in the production and composition of milk [12,30], BCS and LW of cows [28]. In this study, the relationship between the period of production and the various performance per cow metrics is likely to be indirectly reflecting physiological changes in cows associated with seasonal calving.

It is likely that seasonal factors including herbage nutritive value, photoperiod, rainfall and weather can confound the effect of changes in the physiology of cows affecting thus cow performance [29,31,32]. However, the effects of a number of seasonal climate, herbage NV and herbage quantity factors were controlled for in the analysis of the data. Consequently, much of the variation in performance per cow explained by period of production could be due to other factors associated with the seasonality of the farm system that were not considered in this study and that might be influencing animal performance (e.g., proportion of legumes in the herbage mix).

### 4.3. Importance of Herbage Nutritive Value and Quantity on Performance per Cow

In agreement with Brun-Lafleur et al. [33] who found significant relationships ($p < 0.05$) between the energy x protein interaction of diets and MY, FY and PY in experimental dairy cows, we found ME and CP, through their interaction effect, were the most relevant NV traits explaining milk production at Dairy 1. It is likely that the ME×CP effects on milk production observed at Dairy 1 are the consequence of an unbalanced diet that for energy and protein. This suggestion is supported by the negative relationship observed between ME and MU (Table 5) and the fact that MU has been described as a useful indicator of appropriateness of the crude protein to energy ratio in the diet of dairy cows [34]. Moreover, the negative relationship between ME and MU also suggests that the imbalance was most likely due to excess CP rather than energy. Other authors [35,36] have also highlighted excess CP in pastures in New Zealand dairy farms.

The mechanisms by which ME×CP relate with milk production are multiple and complex. Varying contents of ME and CP in herbage signify variation in the availability of these nutrients for metabolic processes per kilogram of dry matter consumed [3]. In addition, ME and CP also affect the total supply of nutrients by influencing DMI [37,38] and, thus, the performance of grazing cows [6]. More importantly, dietary protein content also alters the efficiency in which other nutrients are absorbed and partitioned toward mammary secretion [39,40]. Opposite to the relationship observed between ME and MU, the positive relationship between NDF and MU can be explained by lower energy contents in the herbage of increasing NDF that results in reduced efficiency in the use of nitrogen and, thus, increased MU.

The area of herbage offered to the cows was the herbage quantity factor that had the highest relative importance for explaining most performance per cow variables including LWC, PP, PFR and PI2. It is likely that measuring and controlling AH has provided the farm manager with a more accurate allocation of herbage than measuring other quantity factors such as HM. This may be because measuring area is more accurate than measuring herbage mass, as seasonal changes in the structure and species composition of mixed swards might have required changes in the estimation of HM with the rapid pasture meter [41].

Different from several authors [6–8] who suggest HA is the most important herbage quantity factor influencing milk production, this study found that HA was of low relative importance in explaining milk production at Dairy 1. Is possible that the HA used at Dairy 1 was high enough to not limit DMI, thus, having little importance in driving cow performance compared to other studies [6–8]. This suggestion is supported by the fact that the mean HA value used at Dairy 1 (29.5 kg DM/cow/day in average) was higher than the 25 kg DM/cow/day and within the 20 to 60 kg DM/cow/day range suggested by Bargo et al. [5] and Pérez-Prieto and Delagarde [8], respectively, to optimize the performance of both cows and pasture. Even though Pérez-Prieto and Delagarde [8] propose higher values for HA, the relationship between HA and milk production is curvilinear, and very low yield responses to HA are observed with HA increasing above 30 kg DM/cow/day.

### 4.4. Importance of Climate on Performance per Cow

The negative relationship between T and BCS and PI2 might be explained by the negative effect heat has on DMI and its indirect consequence on the mobilization of body energy reserves. When cows are heat stressed, heat gained exceeds heat lost triggering physiological, anatomical or behavioral changes in the attempt to maintain heat balance [9], being reduced DMI, a well-known mechanism by which cows control heat stress [19]. Reduced DMI could then reflect mobilization of energy from body condition to milk production, explaining, thus, the negative relationship between T and BCS and the lack of relationships found between heat-related metrics and milk production traits.

The lack of significant relationships ($p > 0.05$) found between THI and milk production and composition contrasts findings reported in other studies [10,42–44]. However, differences in the results between this and previous studies might be due to methodological or data accuracy differences. For instance, the studies by Bryant et al. [10] and Bernabucci

et al. [42] used data from multiple locations over lengthy periods of time (about 10 years), and Bernabucci et al. [42] presented experimental data collected under controlled conditions of T and humidity. In contrast, our study used data collected from a single farm over a relatively short period of time (two production seasons) with no control over climate variables. Moreover, differences in the accuracies of temperature and, most likely, humidity data that are used in the calculation of THI might also explain differences in the results.

### 4.5. Some Implications to Farm Management

Even though seasonality was a major factor explaining performance per cow at Dairy 1, discussing the implications of altering seasonality is of little practical value, as seasonality is the backbone of the cost leadership strategy used in most farms in New Zealand [29,45]. The major implication of the findings reported in this study relate to the potential of rapid herbage ME and CP measurement for its use in the development of more balanced diets for these nutrients, potentially resulting in more efficient grazing, feed use and reduced risk excess N could have on cows [46,47] and the environment [48]. This can be done, for instance, by replacing herbage of high CP with maize or cereal silage of low CP content [48,49]. However, balancing diets would also require accurate estimates of both nutrients in supplementary feed and nutritional requirements of the cows. Finally, the diet offered to the herd could also include the addition of yeast [50,51] and clay and yeast supplements [52] to take into consideration the negative effect temperature had on cow performance.

### 4.6. Limitations of This Study

The MLR modelling approach assumes no correlation among explanatory variables. However, explanatory climate variables such as temperature and herbage NV are not mutually exclusive, resulting in a violation of the assumption of independence. Violating this assumption can lead to biased and inconsistent estimates of regression coefficients and, therefore, inaccurate interpretation of the results presented in this study. For example, results from MLR models developed to explain the effect of climate variables on herbage NV show that variation in climate can explain between 13 and 39% of the variation in herbage NV (Table A1, Appendix A).

### 5. Conclusions

This study provided field-based empirical evidence showing that the NV of herbage offered daily to cows is of higher relative importance than climate or herbage quantity factors in driving performance per cow in a pasture-based dairy farm. The quantity of herbage supplied at the study farm might have been high enough to not limit cow performance. Rapid herbage NV measurement can be potentially useful to inform decision making around the development of feeding strategies aimed at better matching daily supply and demand of nutrients to improve the feeding efficiency and the overall performance of farms. Further research is required to investigate the extent at which daily variation in herbage NV is able to satisfy the requirements of individual grazing cows.

**Author Contributions:** Conceptualization, F.D., N.L.-V. and S.M.; methodology, F.D. and N.L.-V.; software, F.D.; formal analysis, F.D. and N.L.-V.; investigation, F.D. and N.L.-V.; resources, N.L.-V., N.S., I.D., I.Y. and S.M.; data curation, F.D.; writing—original draft preparation, F.D.; writing—review and editing, N.L.-V., N.S., I.D., I.Y. and S.M.; supervision, N.L.-V., N.S., I.D., I.Y. and S.M.; project administration, N.L.-V., N.S., I.Y. and S.M; funding acquisition, N.L.-V., N.S., I.Y. and S.M. All authors have read and agreed to the published version of the manuscript.

**Funding:** This research received no external funding.

**Institutional Review Board Statement:** Not applicable.

**Informed Consent Statement:** Not applicable.

**Data Availability Statement:** The data presented in this study are available on request from the corresponding author.

**Acknowledgments:** The authors would like to thank Jolanda Amoore, farm manager at Dairy 1 farm, Massey University for facilitating and providing the data used in this study and Eduardo Sandoval-Cruz for his help collecting hyperspectral data.

**Conflicts of Interest:** The authors declare no conflict of interest.

## Appendix A

Multiple linear regression models were developed to explain the effect of climate variables on herbage NV using the modelling approach described in Section 2.4. Variation in climate explained between 13 and 39% of the variation in herbage NV (Table A1). Temperature was the most significant ($p < 0.001$) climate variable explaining all herbage NV traits except CP, which was significantly explained by THI ($p < 0.001$), CSI and Rain ($p < 0.1$).

**Table A1.** Regression coefficients of climate variables explaining herbage nutritive value traits.

| Coefficient | ME | CP | NDF | DM |
|---|---|---|---|---|
| Intercept | 12.03 *** | 32.4 *** | 31.9 *** | 15.0 *** |
| T | −0.08 *** | − | 0.52 *** | 0.45 *** |
| THI | − | −0.11 *** | − | − |
| CSI | − | −0.01 † | − | − |
| Rain | − | 0.10 † | − | − |
| $R^2$ | 0.33 | 0.13 | 0.39 | 0.15 |

† significant at $p < 0.1$, *** significant at $p < 0.001$. ME = herbage metabolizable energy, CP = herbage crude protein, NDF = neutral detergent fiber, DM = herbage dry matter, T = mean daily temperature, THI = temperature humidity index, CSI = cold stress index, Rain = rainfall.

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
