# Peer review of "The Relative Importance of Herbage Nutritive Value and Climate in Determining Daily Performance per Cow in a Pasture-Based Dairy Farm"

_agriculture, doi:10.3390/agriculture11050444_

Round 1
Reviewer 1 Report
The manuscript agriculture-1169488 has a good scientific quality and presents interesting information on major determinants of dairy cattle performance on pasture. Methodologies are adequate to achieve the proposed objectives and are properly described, although some detailed explanation may be missing. Results in general are clearly and succinctly exposed, but some wrong sentences need correction.
One of the most relevant results refers to the higher relative importance of herbage NV than herbage quantity in explaining cow performance. However, as explained in L353-355, it could be due to the fact that herbage allowance during the study was high enough to not limit DMI. This is a key aspect that should be taken into account when comparing the relative influence of herbage NV and quantity parameters driving animal production traits, and so, it should be acknowledged in both the Abstract and Conclusions sections.
Below there are some specific points that need to be corrected or clarified; most of them refer to minor formatting and grammar issues.
L17: Please add the country.
L73: As 'OAD' is no longer repeated throughout the manuscript, the abbreviation can be removed.
2.2.2. Somewhere in the subsection the timing and frequency of the samplings performed to determine herbage NV should be added. It is not clear if these samplings were performed coinciding with herbage mass measurements.
L162-164: According to this rescaling of PCA axes, should the scores for original variables (loadings in Table 2) be also rescaled to 0-100 ranges?
L202-203: Yields for urea is repeating MUY.
L203: The maximum CV% value of 68.2 does not appear in Table 3. Should it be changed to 47.9 (PI2)?
L2070-271: There is something wrong in the sentence. I think that, instead of ME and BCS, the authors are referring to NDF and MU. Nevertheless, the sign of the relationships seem incorrect according to Table 5. In my opinion, it should read as follows: "being the relationship between ME and MU and between NDF and MU negative and positive, respectively (P < 0.01)." Please check it.
L273: Change P-value of ME-BCS relationship to P < 0.05, as shown in Table 5.
L309: Delete repeated 'that'.
L316: Change the title of subsection 4.3.
L321-322: There is something missing in the sentence, or should 'that' be deleted?
L325: suggests
L331: affect
L353: It is possible that…
L388: The reference [51] does not seem related to environmental effects, but rather to effects on cows.
L393: The references [53,54] do not deal with clay supplementation, but with yeast. Should the sentence refer to both clay and yeast supplements?
L426: …; Muller, L.D.; Kolver, E.S.; Delahoy, J.E.
L435: Article title should be in lowercase letters, also for references 21, 22, 25 and 36.
LL442: Dunshea, F.R.; (add period to initial R)
L444: O’Donovan, M.;
L445: Change journal name 'Animals' to 'Animal'.
L448: Write "in situ" in italics.
L468: Book Titles with initial uppercase letters for principal names (also in [50]), and in italics. Scott Foresman: (change the comma to a colon behind editorial name, also for [41]).
L496: Add missing year (1987).
L503: Rattray,
L524: Loganathan, P., (add a blank before initial)
L531: Add the country.
L533-534, 539: Write "Saccharomyces cerevisiae" in italics.
L540: aflatoxin B1. Write "1" as a subscript.
Author Response
Please find attached responses to Reviewer 1.

Reviewer 2 Report
Dear Authors,
Your manuscript was easy to read and presents some very useful information for low-input pasture-based dairy systems. Below are a few questions/comments to improve your manuscript.
Line 83 -84: Stocking rate was define based on the total area. What about stocking density for the area grazed at that particular period? Was the area grazed per period/day uniform in size?
I have not seen a clear experimental design/approach to account for variation at the paddock level.
Line: 107-109: Is this the best way to calculate herbage allowance? herbage mass/number of animals? Why not the total liveweight on pasture? In other words why the total animal liveweight was not included in the calculation of HA?
Line 158: What variables are in question here is it from table 1? If so, then cite table 1 after variables (Table 1).
Conclusion: A comment to ponder on. Is climate mutually exclusive from forage nutritive value? If not how do you account for the influence of climate on forage nutritive value over time and the combined influence on the performance of dairy cows?
Once these comments can be clarified, it will improve the quality of the manuscript.
Author Response
Please find attached responses to Reviewer 2.

Reviewer 3 Report
The paper gives interesting information about the relative importance of different factors that influence the productivity of dairy cows under a seasonal pasture based system as practiced in New Zealand. It challenges the concept that it is mainly the herbage allowance that determines the productivity of the dairy cows in this system and thus gives way to new thoughts and ideas on what knowledge is needed to further improve dairy production at the farm level, mainly the importance of assessment of pasture nutritive value as a future management tool. However, there are questions and concerns in the methodology that need to be addressed.
General comment:
The paper gives interesting information about the relative importance of different factors that influence the productivity of dairy cows under a seasonal pasture based system as practiced in New Zealand. It challenges the concept that it is mainly the herbage allowance that determines the productivity of the dairy cows in this system and thus gives way to new thoughts and ideas on what knowledge is needed to further improve dairy production at the farm level, mainly the importance of assessment of pasture nutritive value as a future management tool.
Method
Lines 96-103: The terms “measurement period” (first introduced on line 96) and grazing event (line 103) are not well defined and explained. Was the same value for herbage mass used for the entire 2-3 week period (as this was measured every two or three weeks)?
What was the basis for each observation? In table 3, N=140 observations. Does that mean that there were approximately 70 observations per season? The observation period lasted approximately from 31 July/1 August one year until 8/9 April the next year. This would mean around 35-36 weeks. Were the measurement periods and grazing events evenly distributed over these periods each year? That would mean around 2 grazing events per week.
How can observations that are collected after each other in time be regarded as independent observations? For instance: The milk yield of the herd on one occasion must be closely related to the milk yield on the next occasion. In a similar manner, many of the other variables must be related from one observation to the next. Perhaps the statistical model accounts for this but, as far as I can understand, the section about the statistics does not really explain how this relation is taken into consideration in the statistical analysis.
Lines 253-256: When it states that the average relative importance of herbage associated variables in explaining any of the performance per cow dependant variable was higher than the average relative importance of the climate variables it would be beneficial to define what variables are included in what is summarized as “herbage associated variables” and what is included in “climate variables”. Does herbage associated variables include all of the following variables: HM, HA, AH, PD, ME, CP, MExCP, NDF and DM? And are the variables T, CSI, and Rain included in the climate variables? In that case, this needs to be defined or perhaps an extra column can be added in table 4 where the term herbage associated variables and climate variables are shown.
Lines 255-257: Furthermore, do you compute an average of the variables ME, CP, MExCP, NDF and DM when you write about the average relative importance of herbage NV traits? In a similar manner, what is included in the herbage quantity associated variables. All the different groups of variables must be clearly defined in the paper.
Lines 257-263. Unfortunately, I do not understand: In the beginning of this section POP explains 2-51% of the variation but at the end of this section, it is stated that the relative importance of this factor is low (0.02%). Obviously, I am lacking in understanding in some way but perhaps you could explain this in a better way? Perhaps with more explanatory text….
Line 338-339. It is stated that: “The negative relationship between NDF and LW may be a consequence of reduced rumen fill resulting from reduced DMI.” It is true that high NDF values leads to a reduced DMI but not necessarily to a reduced rumen fill. The passage rate is reduced with high NDF values, this is certainly true, but the rumen fill can be high due to a low digestibility. The herbage remains for a long time in the rumen when it has low digestibility. The rumen can be full but intake and passage rate will be low with high NDF values. Please modify the statement.
On lines 391-393 it is stated: ”Finally, the diet offered to the herd could also include the addition of clay supplements to take into consideration the negative effect temperature had on cow performance (53, 54, 55).” However, only one of these references (no 53, Zhu et al. 2016. Effects of supplement levels of Saccharomyces cerevisiae fermentation products on lactation performance in dairy cows under heat stress. AJAS 29, 801-806) have studied supplements to dairy cows under heat stress and it was not clay but Saccharomyces cerevisiae fermentation products. This paper does not even mention clay and the other papers have not studied heat stress. Please revise text and references, or perhaps just exclude the lines 391-393 and the associated references.
Author Response
Please find attached responses to Reviewer 3.

Round 2
Reviewer 1 Report
The manuscript agriculture-1169488 has been correctly improved. The amendments satisfactorily resolve the issues specified by the reviewers.
Only a few grammatical and editing errors remain to be corrected, in addition to checking the new table.
L124: Change 'was' to 'were'
In equation (3), the minus symbols of the last addend should be represented by long dashes, just like in (40 – T).
Throughout the manuscript, I recommend changing the symbol x for interactions (MExCP) with × as used in equations.
Table 1: The units for AH should be ha/day.
L460: Should 'which ' be added after 'CP, '? The last sentence could be changed to "which was mostly explained by THI (P < 0.001)." since CSI and Rain are marginally significant (P < 0.1). Regarding this, it is somewhat surprising that the coefficient for THI (-0.11) is highly significant, whereas that for Rain, being just a litte lower in absolute terms (0.10), is much less significant. Please check the coefficient values.
L485: Article title should be in lowercase letters.
L519: package
L578, 584-585: Book Titles with initial uppercase letters for principal names.
Author Response
Please find document attached
Reviewer 2 Report
Dear Authors,
Thanks for addressing all my comments.
Author Response
Dear Reviewer,
Thank you for your valuable input and time put into the revision.
The Authors